# 3D-3D Superimposition of Pubic Bones: Expanding the Anthropological Toolkit for the Pair-Matching of Commingled Skeletal Remains

**DOI:** 10.3390/biology12010030

**Published:** 2022-12-23

**Authors:** Andrea Palamenghi, Annalisa Cappella, Michaela Cellina, Debora Mazzarelli, Danilo De Angelis, Chiarella Sforza, Cristina Cattaneo, Daniele Gibelli

**Affiliations:** 1LAFAS (Laboratorio di Anatomia Funzionale dell’Apparato Stomatognatico), Dipartimento di Scienze Biomediche per la Salute, Università degli Studi di Milano, Via L. Mangiagalli 31, 20133 Milan, Italy; 2LABANOF (Laboratorio di Antropologia e Odontologia Forense), Sezione di Medicina Legale, Dipartimento di Scienze Biomediche per la Salute, Università degli Studi di Milano, Via L. Mangiagalli 37, 20133 Milan, Italy; 3U.O. Laboratorio di Morfologia Umana Applicata, IRCCS Policlinico San Donato, 20097 San Donato Milanese, Italy; 4Dipartimento di Scienze Biomediche per la Salute, Università degli Studi di Milano, Via L. Mangiagalli 31, 20133 Milan, Italy; 5Reparto Di Radiologia, Ospedale Fatebenefratelli, ASST Fatebenefratelli Sacco, 20121 Milan, Italy

**Keywords:** commingled remains, pubic bones, point-to-point distance, virtual anthropology

## Abstract

**Simple Summary:**

The resolution of commingled assemblages is a highly demanding task in forensic anthropology, where intermixed skeletal remains from several individuals have to be sorted to their respective persons of origin. The issue has been addressed with morphological and osteometric analyses and, more recently, with virtual techniques. Digital superimposition of bone models harmonizes with this research avenue by providing a quantification of the similarity between bones that can be used for pair-matching (i.e., association of right and left bones) analysis. This study considers the pair-matching of pubic bones by superimposing three-dimensional (3D) models of bones acquired through computed tomography (CT). The correct sorting of commingled pubic bones (and, therefore, innominate bones) is paramount for the creation of accurate biological profiles of the remains. The point-to-point distance (in mm) analysis, resulting from the left-on-right superimposition of pubic bone models, allows for the determination of a threshold that discriminates pairs of pubic bones that belong to the same individual (match) from those of different individuals (mismatch). This study thus contributes to expanding the tools available to forensic anthropologists tasked with the pair-matching of bones, which is specifically relevant for the resolution of mass disasters where commingled skeletal remains are recovered.

**Abstract:**

Virtual anthropology (VA) has recently produced an additional tool for the analysis of commingled remains and is based on the distance analysis between three-dimensional (3D) models of bones. To date, the pair-matching of the innominate bone through a 3D approach remains partially unexplored. Here, 44 abdominal CT scans (22 males and 22 females) were selected from a hospital database, and the pubic bones were segmented through ITK-SNAP software. The models were hollowed with Viewbox4 to minimize the amount of trabecular bone. The left pubic bones were mirrored and superimposed on the right ones, according to the smallest point-to-point difference between the two surfaces through VAM software. RMS distances between models were calculated through VAM, producing RMS values for 20 matches and 420 mismatches for each sex group. Differences in RMS distance values between matches and mismatches were investigated through Mann–Whitney tests (*p* < 0.05); the repeatability of the procedure was assessed through absolute and relative technical error measurement (TEM and rTEM). RMS distance values of matches and mismatches were significantly different (*p* < 0.01) in both groups. The method yielded optimal results with high sensitivity (100.0%) and specificity (99.8% in males, 98.8% in females) rates according to the chosen threshold. This project contributes to the research field of VA with a valuable adjunct that may bolster and strengthen the results of the current visual and osteometric methods through a multidisciplinary approach.

## 1. Introduction

When analyzing commingled assemblages, the pair-matching of bilateral (i.e., left and right) bones represents one of the first steps of the unmingling procedures. The methods traditionally rely on a visual examination of the similarity according to bone size and shape [1]. However, these are based upon subjective observations which entail decreased efficiency when the commingled assemblage is complicated by higher numbers and similar aspects of the recovered remains [2]. Osteometric statistical methods assessing similar metric features between pair and symmetric bones provide objective tools, although they are still restrained by the reference skeletal populations from which they were developed [3,4]. It is, therefore, worth exploring new pathways that may help to resolve the above-mentioned issues of the current practices traditionally applied to pair-matching. Within the field of forensic anthropology, the analysis of bones through innovative imaging visual aids has recently been expanding, and virtual anthropology (VA) is now flourishing with new studies praising the use of digital bone models for various analyses [5,6,7]. The possibility of isolating three-dimensional (3D) models of skeletal remains brought about a novel tool in different subfields of forensic anthropology [8,9,10,11,12], including the investigation of commingled bones as well. In this regard, a method for the pair-matching of right and left humeri was engineered, including both a manual and automated version [13]. The mesh-to-mesh value comparison (MVC) method is based on the superimposition of 3D bone surfaces and the calculation of the point-to-point distances between models to predict whether they belong to the same individual. The MVC method showed to be successful independent of the sex, population, and secular trend of the bones. Thus, it represents a powerful implement that could considerably help in sorting the commingled remains of a mass disaster, for example, thus reducing the number of samples to be tested through DNA analysis. At present, the digital pair-matching of temporal bones [14], phalanges [15], and clavicles [16] has been investigated and suggest the promising potential of the technique. The 3D superimposition of bone models was applied also to the re-association of articulating elements, such as the temporomandibular [17] and the atlantooccipital joints [18]. 

To date, the performance and efficiency of the digital pair-matching method on other bones are still unexplored, and the innominate bone is among these. Recently, a test on the digital pair-matching of iliac bones affected by postmortem damage recorded lower accuracy rates than previous studies [19]. As this drawback seems to affect the reliability of the method, this new pilot study extends this research line by investigating the possible application of 3D pair-matching to another anatomical region: the pubic bones. These are reliable indicators for sex [20,21,22] and age-at-death [23,24] estimation. In a commingled setting, it is, therefore, paramount to correctly discern between pairs of pubic bones (and, if not fragmented, of the innominate bones) to create an accurate biological profile of the individual represented by the assemblage. The results may represent a novel contribution to the sensitive and dateless topic of commingled remains.

## 2. Materials and Methods

For this study, 44 unenhanced abdominal CT scans of 22 male and 22 female individuals were randomly selected from a hospital database. All CT scans were acquired using the same CT equipment (Somatom Definition Flash, Siemens, Engelberg, Germany), with the following acquisition parameters: kVp 120, reference mAs 200, with automated current modulation, collimation 128 × 0.6 mm, pitch 2, reconstructed with a slice thickness of 0.75 mm with a bone reconstruction algorithm. The examinations were performed as screening visits for renal colic, abdominal and inguinal hernia. Males’ ages were between 22 and 58 years (mean age: 41.3 ± 9.2 years), whereas females were aged between 29 and 56 years (mean age: 43.6 ± 8.3 years). The data collection was approved by the local ethical committee (7331/2019) and followed international ethical guidelines (the Helsinki protocol). All the chosen individuals were free from congenital and acquired pathologies involving the innominate bone.

### Acquisition and Superimposition Protocol

The CT scan images of each subject were loaded on ITK-SNAP [25], which allows for the isolation of anatomical structures and the production of accurate 3D models through a process called segmentation. The region of interest (ROI) was set at the pubic bones, choosing the medial edge of the iliopubic eminence and the inferior edge of the ischiopubic ramus as the lateral and inferior limits, respectively. The semi-automatic segmentation works as follows: seeds or “bubbles” permeating the space of the ROI according to homogeneous grey levels are inserted in the model, resulting in a replica of the anatomical structure. The 3D models of pubic bones generated via segmentation were saved in the STereoLithography interface format (.stl). Each pubic bone model was then individually loaded on the software Viewbox 4 (dHAL software, Kifissia, Greece) to remove any remnants of the inner trabecular bone and leave only the outer shell of the model. The models were thus hollowed, using the “select by visibility” function on Viewbox 4, which automatically selected the inner portions of the model and then deleted them through the function “delete selection”. The spongious bone was removed, as it may influence the following point-to-point alignment of models and alter the correct anatomical superimposition.

The models were then elaborated through the Vectra Analysis Module (VAM) software (version 2.8.3, Canfield Scientific Inc.) to isolate the 20 left and 20 right pubic bone models from the same subject. In addition, four left models (two from each sex group) were acquired in order to include unpaired pubic bones. A precise protocol for the superimposition of the models was designed, in view of the fact that variable results can be obtained according to the reference model [26]. In the present study, the right pubic bone was chosen as the reference model onto which the left pubic bone would be moved. Each left pubic bone was then mirrored and registered onto the right pubic bone of the same and different subjects. Three landmarks were positioned on the left and right models at the most superior and most inferior points of the pubic symphysis and the most medial point of the obturator foramen. This was only to direct a rough alignment of models according to the smallest distance between corresponding landmarks on the two models. Following this, a fine registration based on the smallest point-to-point distance between all the points of one surface mesh to the other according to the whole surfaces was automatically performed by the VAM software. Once the models were aligned, the root mean square (RMS) point-to-point distance (in millimeters) of the left model according to the right one was calculated through VAM, which allows for the visualization of the distance between models also as a chromatographic map (Figure 1). The RMS is the square root of the mean of the squared distances of each point of the model. This is a more reliable proxy for the evaluation of differences and similarities between virtual models, since, in this way, negative and positive distances do not elide each other as they would when only considering arithmetic mean [11,19]. Here, only the models’ shape was considered, and no scaling was performed, following the methodologies of previous works [13,14,15,16,17,18,19].

The sorting performance of the method was evaluated through sensitivity (i.e., the proportion correctly identified as true pair matches) and specificity rates (i.e., the proportion correctly identified as pair mismatches), by arbitrarily choosing an RMS threshold. The procedures of segmentation and superimposition were performed by two operators, and repeatability was assessed using the models of 10 subjects through the calculation of intra- and inter-observer error expressed as the absolute and relative technical error of measurement (TEM and rTEM, respectively). Possible differences between the RMS values of matches and mismatches were evaluated through Mann–Whitney testing separately for males and females (*p* < 0.05).

## 3. Results

A total of 880 superimpositions were performed, resulting in 40 matches and 840 mismatches, with corresponding RMS distance values (shown in Figure 2 and Figure 3) equally divided between males and females. The procedure proved highly repeatable (Table 1); the intra-observer and inter-observer relative technical error measurement (rTEM) values were 4.3% and 5.5% for matches, and 3.0% and 5.7% for mismatches, respectively. The agreement between the two observers and different observations from the same observer could be ranked as “good” [27].

In males, the RMS distance values of matches ranged between 0.41 mm and 0.91 mm (mean: 0.71 ± 0.11 mm), whereas RMS distance values of mismatches ranged between 0.87 mm and 5.63 mm (mean: 2.10 ± 0.81 mm). In females, RMS distance values of matches were between 0.47 mm and 0.77 mm (mean: 0.68 ± 0.13 mm), and between 0.77 mm and 3.49 mm (mean: 1.68 ± 0.47 mm) in the set of mismatches. According to a Mann–Whitney test, the RMS distances of matches were significantly different (lower) than mismatches (*p* < 0.01) for both males and females. Table 2 summarizes the sorting performance of the method for the two groups. Among males, when the RMS threshold was arbitrarily set at 0.91 mm, the method was 100.0% sensitive and identified all 20 true pair matches, including one pair that was a false positive value (99.8% specific). Among females, the threshold value of 0.89 mm recognized all the true pairs (100.0% sensitive), although five false positive pairs were identified as well, hence the lower specificity rate (98.8%). Overall, when pooling together the two sexes, the use of 0.91 mm as a threshold was 100.0% sensitive and 99.3% specific. Although the thresholds produced false positive values, the RMS distance values of the true pair matches were always the lowest values across all of the comparisons.

## 4. Discussion

The unmingling process is a highly demanding task in anthropology, which requires sound methodologies. Thanks to the most recent advances in technology, anthropologists benefit from quantifying morphological similarities between bones using 3D osteological models [18,28]. The limitations of visual and osteometric methods for the pair-matching of commingled bones opened the path to the investigation of a novel three-dimensional (3D) approach. Using 3D-3D superimposition for unmingling purposes is, therefore, currently expanding in anthropological research [13,14,15,16,17,18,19], demonstrating researchers’ great interest in the topic, possibly because this tool represents a powerful addition to traditional methods.

This study on the pair-matching of pubic bones fits within this novel research field with promising results. A previous attempt to quantify the similarity of right and left innominate bones for pair-matching took into consideration iliac bones and recorded low accuracy rates. Therefore, the analysis of pubic bones offered data on a skeletal region that could produce more reliable results to help anthropologists with the sorting of innominate bones. Indeed, the sorting performance when using the threshold value is high, as the method can efficiently distinguish the true pairs of pubic bones from the false pairs, except for one (in males) and five (in females) false positives that generated RMS distance values below the cut-off. However, in cases of commingled remains, false positive values are more acceptable than false negatives, as they can be further analyzed through other methods to confirm the assessment. In contrast, false negative values would deem a true pair as a mismatch, which would be eventually discarded [19]. Despite the sorting performance being flawed by these false positive values, the outcomes of the method are concordant with those reported by previous studies using the manual superimposition of 3D bone models for digital pair-matching (Table 3). In detail, in comparison with the sorting performance of the pair-matching of humeri [13] and temporal bones [14], pubic bones showed slightly lower specificity rates. It is to be pointed out that, although the acquisition technique was the same (CT scan), the parameters of acquisition are different, and the proprietary algorithms used to calculate point-to-point distances may reasonably differ according to the software used [13].

This study is based on skeletal portions limited to the pubic bone, so it is possibly suitable for fragmented remains presenting this specific anatomical structure. As much as with visual and osteometric methods, incompleteness and fragmentation may reasonably represent a drawback for digital pair-matching as well. Previous tests on artificially generated fragments of clavicles [16] yielded sensitivity rates ranging from 81.3% to 87.6%, according to the type of fragment (i.e., acromial, midshaft, and sternal). Another study on iliac bones from a cemeterial collection [19] recorded a specificity rate of 51%, thus contributing to this topic with preliminary evidence. So far, the issue of taphonomic condition and its influence on 3D superimposition may be considered a limitation of the method. As such, the application of the digital pair-matching of fragmented bone portions currently requires further investigation, as it seems that it may be affected by the preservation state of the remains [16,19]. Future studies should thus include bones with different taphonomic assets, in order to thoroughly evaluate how digital pair-matching performs according to the preservation state.

Only automated superimposition was investigated on clavicles [16]. Although automation could save time, especially when large numbers of bones are examined, the related performance on complete bones was less optimal than results obtained through manual superimposition, yielding a maximum of 95.0% sensitivity [13] and 90.9% specificity rates [16]. The limited success of automatic systems may be due to a lack of operators who may check out the suggested superimposition and verify whether they are not correctly performed according to the anatomy and morphology of the superimposed structure. Therefore, although manual superimposition is still time-consuming, it allows the operator to verify how the models are superimposed. Furthermore, the procedure is very straightforward and repeatable, so it can be performed by operators with various levels of expertise. Nonetheless, the results of this study are wedded to computed tomography (CT) as the means of acquisition, which may limit widespread use because of cost, personnel, maintenance, and acquisition time [29]. However, the use of CT scans in forensic anthropology has been expanding both for research and case-work purposes [30,31], and forensic laboratories may be equipped with in-house devices or partner with medical institutes that are equipped with CT units [32,33]. Possibly, future advancements in this field will make such an apparatus more cost-efficient, which may lead to wider investigation and application of the digital pair-matching method.

To the best of our knowledge, this is the first study on 3D pair-matching analyzing pubic bones from the two sexes separately, whereas McWirther et al. [16] pooled together the clavicles of males and females. This seems reasonable since sex estimation on the clavicle would be challenging. When analyzing pubic bones, however, a preliminary sex assessment through well-established and reliable methods [20,22,34] may prove useful to 3D pair-matching for two reasons. Firstly, differences in size between male and female innominate bones may influence the RMS values. Moreover, sex estimation would help reduce the number of possible comparisons. Besides, the suggested thresholds yielded a sensitivity of 100.0% without any false negatives (i.e., correct matches wrongly diagnosed as mismatches), with an overall specificity of 99.3%, which indicates an efficient performance of the method in sorting true and false pairs.

This study provided evidence on the 3D pair-matching of innominate bones, testing males and females separately. Further tests on larger samples would provide insight into the possible use of the method in real-case scenarios. Moreover, as the current sorting techniques are often hampered by homogenous assemblages in terms of sex, age, and size disparities between individuals [2,3,4], further studies are needed to verify the influence of the above-mentioned parameters on the 3D pair-matching methods currently available in the literature. The main advantage of 3D-3D superimposition relies on the numerical value provided by the comparison, which quantifies the similarity between two pubic bones and can be used as a proxy to segregate and re-associate pairs of innominate bones, thus aiding the resolution of commingled scenarios. However, traditional sorting methods, based on metrical and morphological analyses, maintain their values and are widely used in anthropological practice in cases of commingled remains. The present study does not aim to replace visual and osteometric comparisons but to explore the possible application of an additional tool to be used together with traditional methods in order to strengthen the results within a multidisciplinary approach. 

## 5. Conclusions

This study explored the 3D pair-matching of pubic bones to expand the techniques which may enable anthropologists to quantify the evidence that commingled pubic bones belong to the same individual. The results are promising, with high sensitivity and specificity rates, and may potentially provide a novel tool that quantifies the similarities between pubic bone models together with traditional methods. Further tests are needed to understand the practical applications to real cases. Furthermore, this study contributes to the flourishing field of virtual anthropology with an addition to the current visual and osteometric methods to assess the pair-matching of innominate bones and portions of pubic bones.

## Figures and Tables

**Figure 1 biology-12-00030-f001:**
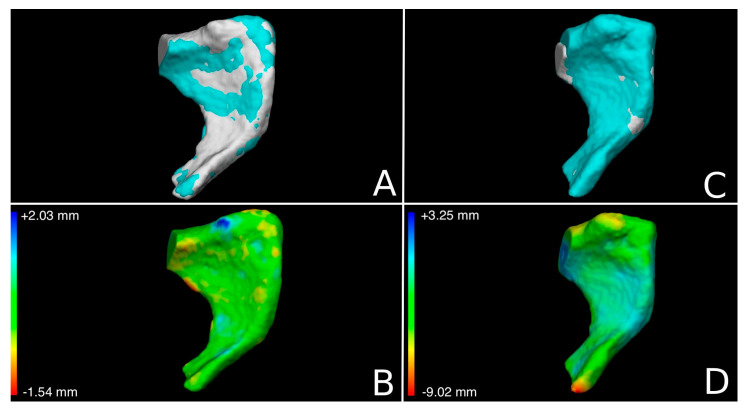
Superimposition and calculation of the point-to-point distance between two models. (**A**,**B**): superimposition left-on-right model from the same subject. (**C**,**D**): superimposition of the left-on-right model from different subjects. (**A**,**C**): the light blue model is the mirrored left pubic bone and the grey one is the right pubic bone. (**B**,**D**): the green areas indicate coincident points on the models. In blue, the receding areas of the left pubic bone according to the right pubic bone. In red, orange, and yellow are the prominent areas of the left pubis according to the right pubis. (**B**): example of the chromatic map of a true pair match. (**D**): example of the chromatic map of a mismatch.

**Figure 2 biology-12-00030-f002:**
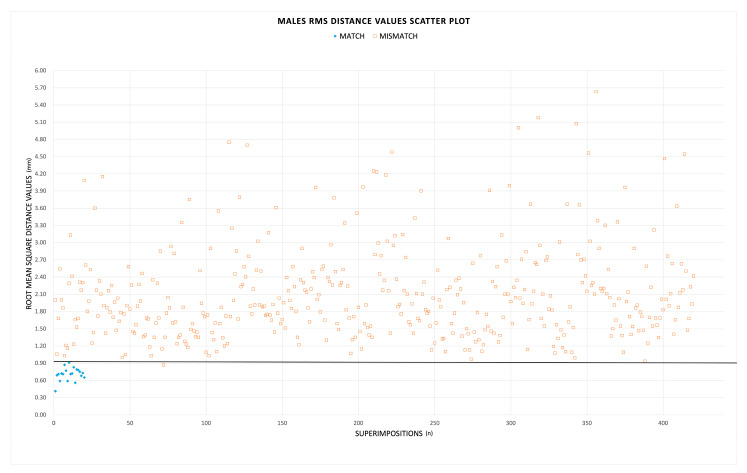
Scatter plot of the RMS distance values for males. The blue diamonds are the 20 RMS distance values of true pair matches, whereas the orange squares are the mismatches. The black line represents the threshold of 0.91 mm.

**Figure 3 biology-12-00030-f003:**
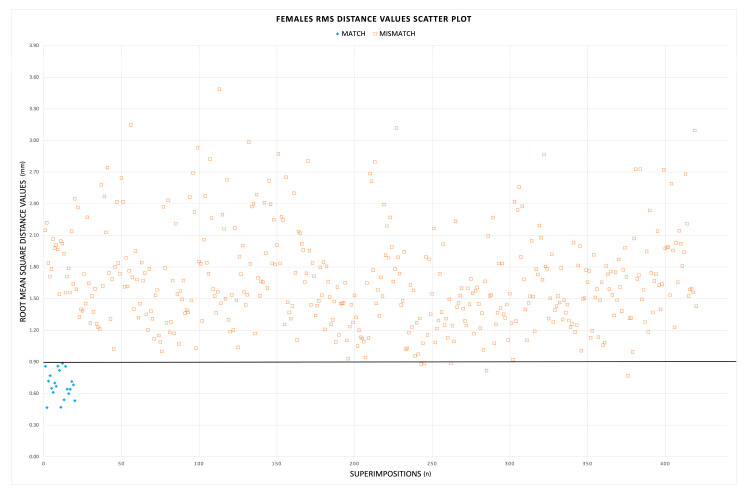
Scatter plot of the RMS distance values for females. The blue diamonds are the 20 RMS distance values of true pair matches, whereas the orange squares are the mismatches. The black line represents the threshold of 0.89 mm.

**Table 1 biology-12-00030-t001:** Intra- and inter-observer error in matches and mismatches. TEM: technical error measurement; rTEM: relative technical error measurement.

Repeatability	Intra-Observer	Repeatability	Intra-Observer
Matches	0.03 mm (4.3%)	0.03 mm (4.3%)	0.03 mm (4.3%)
Mismatches	0.04 mm (5.5%)	0.04 mm (5.5%)	0.04 mm (5.5%)

**Table 2 biology-12-00030-t002:** Sorting performance of the method according to the chosen threshold.

Group	RMS Threshold	True Positive Pairs	False Positive Pairs	True Negative Pairs	False Negative Pairs	Sensitivity	Specificity
Males	0.91 mm	20	1	419	0	100.0%	99.8%
Females	0.89 mm	20	5	415	0	100.0%	98.8%
Combined	0.91 mm	40	6	834	0	100.0%	99.3%

**Table 3 biology-12-00030-t003:** Summary of the sorting performance of digital manual pair-matching across published studies.

Performance				
MANUAL SUPERIMPOSITION	Humeri [13]	Temporal bones [14]	Iliac bones [19]	Pubic bones (Current study)
Sensitivity	100.0%	100.0%	100.0%	100.0%
Specificity	100.0%	100.0%	51.0%	99.3%

## Data Availability

Not applicable.

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
