# Peer review of "3D-3D Superimposition of Pubic Bones: Expanding the Anthropological Toolkit for the Pair-Matching of Commingled Skeletal Remains"

_biology, 2022, doi:10.3390/biology12010030_

Round 1
Reviewer 1 Report
This paper expands on previous methodology of using virtual techniques and superimposition to march pair elements of the human skeleton for purposes of indivisualisation and identification in Forensic Anthropology. This is an important topic with very few original and validation studies and there is definetely scope in engaging in this this research field.
A major methodological issue in this case would be the lack of unpaired bones in the sample. Meaning if lets say 5 single pubic bones (without pair in the sample) are superimposed with the rest, would that change the matching process? Karell et al. (2016) used single bones (humeri) for that particular reason. I would therefore suggest to add 3-5 single pubic bones and repeat the analysis to show a more realistic model and to estimate sensitivity and specificity in a more accurate manner. It is very unlikely in a commingled situation that all bones will definetely have a match.
Other than the above comment the paper is well written and appropriate for publication
Reviewer 2 Report
This paper presents a novel approach to pair matching in order to sort co-mingled remains. Although it is a pilot study, it points to new directions this research into this intractable problem might take. I found it thorough, clearly presented and fully worthy of publication. I have just a minor comment on methodology:
P. 3. I think a bit more explanation of the registration and RMS measurements would be helpful to readers. How many landmarks are involved in the RMS distances? I assume RMS distances are just the square root of the sum of squared distances between all pairs of landmarks? Or the mean? Authors should also clarify whether the comparisons are based on shape alone, or whether size was retained.
Author Response
We thank the reviewer for raising this issue, so we can now clarify.
Landmarks were positioned only to direct a rough alignment of the models. Following this, a fine registration based on the least point-to-point distance between all the points of one surface mesh to the other according to the whole surfaces was automatically performed by the VAM software. Once the models were aligned, the Root Mean Square (RMS) point-to-point distance (in millimeters) of the left model according to the right one was calculated through VAM. The RMS is the square root of the mean of the squared distances of each point of the model. This is a more reliable proxy to evaluate differences and similarities between virtual models, since, in this way, negative and positive distances do not elide each other, as they would do when only considering arithmetic mean [1, 3]. Here, only the models’ shape was considered, and no scaling was performed, following previous works [2, 3].
References
- Gibelli, D.; Cellina, M.; Cappella, A.; Gibelli, S.; Panzeri, M. M.; Oliva, A. G.; Termine, G.; De Angelis, D.; Cattaneo, C.; & Sforza, C. An innovative 3D-3D superimposition for assessing anatomical uniqueness of frontal sinuses through segmentation on CT scans, Int J Legal Med, 2019, 133, 4, pp. 1159–1165, doi: 10.1007/s00414-018-1895-4.
- Karell, M. A., Langstaff, H. K., Halazonetis, D. J., Minghetti, C., Frelat, M., & Kranioti, E. F.A novel method for pair-matching using three-dimensional digital models of bone: mesh-to-mesh value comparison, Int J Legal Med, 2016, vol. 130, no. 5, pp. 1315–1322, doi:10.1007/s00414-016-1334-3
- Palamenghi, A.; Mazzarelli, D.; Cappella, A.; De Angelis, D.; Sforza, C.; Cattaneo, C.; Gibelli, D., Digital pair- matching of iliac bones : pilot study on a three-dimensional approach with models acquired through stereophotogrammetry, Int J Legal Med, 2022, doi: 10.1007/s00414-022-02895-x
Reviewer 3 Report
Dear Editor of Biology and Authors,
I have carefully read the paper “3D-3D superimposition of pubic bones: expanding the anthropological toolkit for the pair-matching of commingled skeletal remains” and believe it to be well-written, theoretically supported by a good bibliographic analysis and scientifically sound. The subject of the paper is utterly relevant for forensic anthropology and bioarcheology, as there are many recovery circumstances where the human bones are commingled. I would only ask the authors to let the readers know if it is possible / feasible to conduct the 3D pairing of the pubic bones even if they are fragmented. That is because the pubic bone is extremely fragile.
Best regards.
Author Response
We thank the reviewer for this relevant point. As much as with visual and osteometric methods, incompleteness and fragmentation may reasonably represent a drawback for the digital pair-matching as well. Previous tests on artificially generated fragments of clavicles [1] yielded sensitivity rates ranging from 81.3% to 87.6%, according to the type of fragment (i.e., acromial, midshaft and sternal). Another study on iliac bones from a cemeterial collection [2] recorded a specificity rate of 51%, contributing to this topic with preliminary evidence. So far, the issue of taphonomic condition and its influence on 3D superimposition may be considered a limitation of the method. As such, the application of the digital pair-matching of fragmented bone portions currently requires further investigation. Future studies should thus include bones with different taphonomic assets, in order to thoroughly evaluate how the performance of the digital pair-matching is affected according to the preservation state.
- McWhirter, Z.; Karell, M.A.; Er, A.; Bozdag, M.; Ekizoglu, O.; Kranioti, E.F. Exploring the Functionality of Mesh-to-Mesh Value Comparison in Pair-Matching and Its Application to Fragmentary Remains. Biology 2021, 10, 1303. https://doi.org/10.3390/biology10121303
- Palamenghi, A.; Mazzarelli, D.; Cappella, A.; De Angelis, D.; Sforza, C.; Cattaneo, C.; Gibelli, D., Digital pair- matching of iliac bones : pilot study on a three-dimensional approach with models acquired through stereophotogrammetry, Int J Legal Med, 2022, doi: 10.1007/s00414-022-02895-x.
Reviewer 4 Report
The paper presents the results of a very interesting study on the use of 3D digital pair matching of pubic bones. Overall I recommend publication, but feel the paper would benefit from minor revision and clarification.
It is unclear whether the authors recommend sorting by sex before applying this method. On the one hand, it is noted that correct sorting is necessary for bio profile estimation. But then the sex-separated data seem to be given preference. The pooled sex data also perform quite well, but only the graphics for male and female RMS distances are presented. Consider adding some discussion on whether you are suggesting that sex be estimated first, or the sorting approach used first to enhance accurate bio profile estimation.
From a practical standpoint, what is the approximate equipment cost and time investment to use this method?
Cases that result in commingling (e.g., mass disasters, mass graves) also often result in bone damage/alteration. Consider adding a discussion on how damaged bone would affect this approach. Do both bones need to be in essentially near-perfect condition in order to apply this method? Are there any other limitations that should be considered?
Author Response
We thank the reviewer for the relevant points.
Point 1: This modified accordingly. We believe that a first triage based on sex estimation would help reducing the number of comparisons. When analyzing pubic bones, a preliminary sex assessment through well-established and reliable methods may prove useful to the 3D pair-matching for two reasons. Firstly, differences in size between male and female innominate bones may influence RMS values. Moreover, sex estimation would help reducing the number of possible comparisons, and the methods for sex estimation from the pubic bones are. Besides, the suggested thresholds yielded a sensitivity of 100.0% without any false negative (i.e., correct matches wrongly diagnosed as mismatch), with an overall specificity of 99.3%, indicating an overall efficient performance of the method in sorting true and false pairs.
Point 2: The use of CT scan in forensic anthropology has been expanding both for research and case-work purposes [1, 2], and forensic laboratories may be equipped with in-house devices, or they partner with medical institutes that are equipped with CT units [3, 4]. Possibly, future advancements in this field will make such an apparatus more cost-efficient, which may lead to a wider investigation and application of the digital pair-matching method.
The machinery used for this retrospective study was leased 10 years ago for 1,2 million euros. The acquisitions can be performed also on cheaper devices, such as multislice CT scans with 16 or 64 detectors. Although the manual superimposition is still time-consuming, it allows the operator to verify how the models are superimposed, whereas automated superimposition recorded lower sensitivity and specificity rates.
Point 3: As much as with visual and osteometric methods, incompleteness and fragmentation may reasonably represent a drawback for the digital pair-matching as well. Previous tests on artificially generated fragments of clavicles [5] yielded sensitivity rates ranging from 81.3% to 87.6%, according to the type of fragment (i.e., acromial, midshaft and sternal). Another study on iliac bones from a cemeterial collection [6] recorded a specificity rate of 51%, contributing to this topic with preliminary evidence. So far, the issue of taphonomic condition and its influence on 3D superimposition may be considered a limitation of the method. As such, the application of the digital pair-matching of fragmented bone portions currently requires further investigation. Future studies should thus include bones with different taphonomic assets, in order to thoroughly evaluate how the performance of the digital pair-matching is affected according to the preservation state.
References
- O’Donnell, C., Iino, M., Mansharan, K., Leditscke, J., & Woodford, N. (2011). Contribution of postmortem multidetector CT scanning to identification of the deceased in a mass disaster: experience gained from the 2009 Victorian bushfires. Forensic science international, 205(1-3), 15-28. doi: 10.1016/j.forsciint.2010.05.026
- Bertoglio, B., Corradin, S., Cappella, A., Mazzarelli, D., Biehler-Gomez, L., Messina, C., Pozzi, G., Sconfienza, L. M., Sardanelli, F., Sforza, C., De Angelis, D., & Cattaneo, C. (2020). Pitfalls of Computed Tomography 3D Reconstruction Models in Cranial Nonmetric Analysis. Journal of forensic sciences, 65(6), 2098–2107. doi:10.1111/1556-4029.14535
- Camine, L. M., Varlet, V., Campana, L., Grabherr, S., & Moghaddam, N. The big puzzle: A critical review of virtual re-association methods for fragmented human remains in a DVI context'. Forensic science international, 2022, 330, 111033. doi: 10.1016/j.forsciint.2021.111033
- Obertová, Z., Leipner, A., Messina, C., Vanzulli, A., Fliss, B., Cattaneo, C., & Sconfienza, L. M. (2019). Postmortem imaging of perimortem skeletal trauma.Forensic Science International, 302, 109921.
- McWhirter, Z.; Karell, M.A.; Er, A.; Bozdag, M.; Ekizoglu, O.; Kranioti, E.F. Exploring the Functionality of Mesh-to-Mesh Value Comparison in Pair-Matching and Its Application to Fragmentary Remains. Biology 2021, 10, 1303. https://doi.org/10.3390/biology10121303
- Palamenghi, A.; Mazzarelli, D.; Cappella, A.; De Angelis, D.; Sforza, C.; Cattaneo, C.; Gibelli, D., Digital pair- matching of iliac bones : pilot study on a three-dimensional approach with models acquired through stereophotogrammetry, Int J Legal Med, 2022, doi: 10.1007/s00414-022-02895-x.